# Chromosome-level genome of *Schistosoma haematobium* underpins genome-wide explorations of molecular variation

**Andreas J. Stroehlein[1], Pasi K. Korhonen[1], V. Vern Lee[2], Stuart A. Ralph[2], Margaret Mentink-Kane[3], Hong You[4], Donald P. McManus[4], Louis-Albert Tchuem Tchuenté[5,6], J. Russell Stothard[6], Parwinder Kaur[7], Olga Dudchenko[8,9], Erez Lieberman Aiden[7,8,9,10,11], Bicheng Yang[12], Huanming Yang[13,14], Aidan M. Emery[15,16], Bonnie L. Webster[15,16], Paul J. Brindley[17], David Rollinson[15,16], Bill C. H. Chang[1], Robin B. Gasser[1]\*, Neil D. Young[1]\***

**1** Faculty of Veterinary and Agricultural Sciences, The University of Melbourne, Parkville, Victoria, Australia, **2** Department of Biochemistry and Pharmacology, Bio21 Molecular Science and Biotechnology Institute, The University of Melbourne, Parkville, Australia, **3** NIH-NIAID Schistosomiasis Resource Center, Biomedical Research Institute, Rockville, Maryland, United States of America, **4** Immunology Department, QIMR Berghofer Medical Research Institute, Brisbane, Queensland, Australia, **5** Faculty of Sciences, University of Yaoundé I, Yaoundé, Cameroon, **6** Department of Parasitology, Liverpool School of Tropical Medicine, Liverpool, United Kingdom, **7** UWA School of Agriculture and Environment, The University of Western Australia, Perth, Western Australia, Australia, **8** The Center for Genome Architecture, Department of Molecular and Human Genetics, Baylor College of Medicine, Houston, Texas, United States of America, **9** Center for Theoretical Biological Physics, Rice University, Houston, Texas, United States of America, **10** Shanghai Institute for Advanced Immunochemical Studies, ShanghaiTech, Pudong, China, **11** Institute of MIT and Harvard, Cambridge, Massachusetts, United States of America, **12** BGI Australia, Oceania, BGI Group, CBCRB Building, Herston, Queensland, Australia, **13** BGI-Shenzhen, Shenzhen, China, **14** Shenzhen Key Laboratory of Unknown Pathogen Identification, BGI-Shenzhen, Shenzhen, China, **15** Parasites and Vectors Division, The Natural History Museum, London, United Kingdom, **16** London Centre for Neglected Tropical Disease Research (LCNTDR), London, United Kingdom, **17** School of Medicine & Health Sciences, Department of Microbiology, Immunology & Tropical Medicine, George Washington University, Washington DC, United States of America

\* robinbg@unimelb.edu.au (RBG); nyoung@unimelb.edu.au (NDY)

**Data Availability Statement:** The sequence data generated here have been deposited in the Sequence Read Archive (SRA) under BioProject

## Abstract

Urogenital schistosomiasis is caused by the blood fluke *Schistosoma haematobium* and is one of the most neglected tropical diseases worldwide, afflicting > 100 million people. It is characterised by granulomata, fibrosis and calcification in urogenital tissues, and can lead to increased susceptibility to HIV/AIDS and squamous cell carcinoma of the bladder. To complement available treatment programs and break the transmission of disease, sound knowledge and understanding of the biology and ecology of *S. haematobium* is required. Hybridisation/introgression events and molecular variation among members of the *S. haematobium*-group might effect important biological and/or disease traits as well as the morbidity of disease and the effectiveness of control programs including mass drug administration. Here we report the first chromosome-contiguous genome for a well-defined laboratory line of this blood fluke. An exploration of this genome using transcriptomic data for all key developmental stages allowed us to refine gene models (including non-coding elements) and annotations, discover 'new' genes and transcription profiles for these stages, likely linked to development and/or pathogenesis. Molecular variation within *S.*

PRJNA78265 (accession numbers SRR15400745-SRR15400772) and PRJNA512907 (DNAzoo repository). Code used for the analysis of data or to create figures is available at https://gitlab.unimelb.edu.au/bioscience/s_haematobium_v3. All other data used are referred to in this article and its supplementary files.

**Funding:** Funding from the Australian Research Council (ARC; LP180101334 to N.D.Y. and P.K.K.; LP180101085 to R.B.G. and B.C.H.C.; and DP160100389 to S.A.R.), BGI and Yourgene Singapore supported this project. The research was supported by the LIEF HPC-GPGPU facility hosted at the University of Melbourne, was established with the assistance of ARC LIEF grant LE170100200. P.K. is supported by the University of Western Australia. E.L.A. was supported by the Welch Foundation (Q-1866), a McNair Medical Institute Scholar Award, an NIH Encyclopedia of DNA Elements Mapping Center Award (UM1HG009375), a US-Israel Binational Science Foundation Award (2019276), the Behavioral Plasticity Research Institute (NSF DBI-2021795), NSF Physics Frontiers Center Award (NSF PHY-2019745), and an NIH CEGS (RM1HG011016-01A1). The funders had no role in study design, data collection and analysis, decision to publish, or preparation of the manuscript.

**Competing interests:** The authors have declared that no competing interests exist.

*haematobium* among some geographical locations in Africa revealed unique genomic 'signatures' that matched species other than *S. haematobium*, indicating the occurrence of introgression events. The present reference genome (designated Shae.V3) and the findings from this study solidly underpin future functional genomic and molecular investigations of *S. haematobium* and accelerate systematic, large-scale population genomics investigations, with a focus on improved and sustained control of urogenital schistosomiasis.

## Author summary

More than 100 million people are infected with the carcinogenic blood fluke *Schistosoma haematobium*, the aetiological agent of urogenital schistosomiasis—a neglected tropical disease (NTD). In spite of its major significance, little is known about this fluke, its interactions with the human and snail intermediate hosts and the pathogenesis of the urogenital form of schistosomiasis at the molecular and biochemical levels. To enable research in these areas, we report the first chromosome-level genome and markedly enhanced gene models for *S. haematobium*. Comparative genomic analyses also reveal evidence of past introgression events between or among closely related schistosome species. This present reference genome for *S. haematobium* and the findings from this study should underpin future functional genomic and molecular investigations of *S. haematobium* and accelerate systematic, large-scale population genomics investigations, with a focus on improved control of urogenital schistosomiasis.

## Introduction

Urogenital schistosomiasis, caused by the blood fluke *Schistosoma haematobium*, is one of the most neglected tropical diseases (NTDs) worldwide, afflicting more than 100 million people, particularly in Africa and the Middle East [1,2]. This disease is transmitted to humans *via* aquatic snails (intermediate hosts) typically of the genus *Bulinus* [3] and is characterised by granulomata, fibrosis and calcification in the urinary bladder wall and other parts of the urogenital tract [4,5], with complications including increased susceptibility to HIV/AIDS [6] and squamous cell carcinoma of the urinary bladder [7,8].

Although no vaccine is available to prevent urogenital schistosomiasis, affected people can be treated with the anthelminthic drug, called praziquantel. However, treatment efficacy with this drug can be variable [9–13], such that mass treatment alone might not achieve a sustainable control of this disease. Effective control is achieved by breaking the transmission of infection/disease, which requires sound knowledge and understanding of the biology and ecology of *S. haematobium*.

A number of studies have shown marked molecular genetic variation within *S. haematobium* [14–17], and some have provided evidence of hybridisation and/or introgression events occurring between members of the *S. haematobium*-group (e.g., *S. haematobium*, *S. bovis*, *S. curassoni*, *S. guineensis*, *S. intercalatum* and *S. mattheei*) in regions of sympatry in continental Africa and, more recently, in France (Corsica) [18–21]. Most studies utilised nuclear ribosomal or mitochondrial DNA, or biochemical markers, and genome-wide investigations are starting to be employed [22]. Thus, it would be highly beneficial to conduct genome-wide analyses of genetic variation within the species currently recognised as *S. haematobium* and closely related species (i.e. of the "*S. haematobium*-group") [23,24].

Central to such expanded analyses will be the availability of a high-quality genome for a well-defined line of *S. haematobium*. Although progress has been made in this direction [25,26], draft genomes for *S. haematobium* remain fragmented and their gene annotations incomplete, compromising comprehensive analyses of molecular variation. Here we use a combination of Hi-C sequencing, and long-read nanopore and PacBio data to produce and annotate the first chromosome-level genome for a well-defined laboratory line of *S. haematobium*, and explore the nature and extent of molecular variation within *S. haematobium* at different stages of development from distinct hosts and from multiple geographic locations in Africa. We discuss the implications of this work for future, large-scale population genetic investigations and for the exploration of hybridisation and introgression events in natural schistosome populations.

## Results

### Reference genome (Shae.V3)

We assembled a chromosome-level reference genome (Shae.V3) for an Egyptian strain of *S. haematobium* [27] from 30,735,883 paired-end Hi-C reads, 4,532,276 Oxford Nanopore long-reads (S1 Table) as well as mate-pair and PacBio data sets (NCBI accession number PRJNA78265) available from previous studies [25,26]. Shae.V3 was assembled into 163 scaffolds, estimated at 400.27 Mb, 98% of which were represented in eight chromosomes. These inferred chromosomes have high synteny to those of *S. mansoni via* 380 linked, syntenic blocks of genes representing 96% of the *S. mansoni* genome (Fig 1A). One of the linked blocks represented a rearrangement between chromosomes 2 and 3 (S2 Table). A comparison of Shae.V3 with assemblies for *S. japonicum* and *S. bovis* (Figs 1B and 1C) showed similar levels of synteny (502 and 339 syntenic blocks, respectively), but a lower percentage of linked scaffolds (83.0% and 33.8%, respectively). For *S. japonicum*, eight rearrangements were evident, whereas rearrangements were not detected for *S. bovis*. A comparison of genome Shae.V3 with a previous assembly for *S. haematobium* (i.e. Shae.V2 with 130 linked scaffolds) [26] (Table 1; Fig 1D) showed that Shae.V3 is substantially more contiguous.

### Gene models and annotation

We transferred 6277 high-confidence gene models (i.e. 67.4%) from Shae.V2 to Shae.V3, and inferred 3154 more genes based on evidence from mapped long and short RNA sequence reads (~ 20.5 million and ~ 277.9 million, respectively) from all key developmental stages and both sexes of *S. haematobium*. All 9431 gene models were encoded on 155 scaffolds, with most (*n* = 9182; 97.4%) located on the eight scaffolds representing all chromosomes of *S. haematobium*.

Of all 9431 genes, 9246 (98%) had orthologs in one or more of three other schistosome species (*S. mansoni*: *n* = 8462; *S. japonicum*: 7953; *S. bovis*: 9246), which clustered into 8370 ortho-groups (S3 Table). All 14,700 isoforms predicted for the 9431 genes were supported by RNA-Seq data (S4 Table). Short and long-read transcript data also provided support for 12,563 5'- and 12,888 3'-untranslated regions (UTRs).

The gene set inferred for Shae.V3 is superior to that of Shae.V2, achieving a higher overall BUSCO score, with fewer fragmented or missing BUSCOs (Table 2). It also contains novel genes, inferred using RNA-Seq evidence for the sporocyst (*n* = 19), cercaria (9), schistosomule (2) stages or eggs from urine (15) (S4 Table). Of all 14,700 conceptually translated protein sequences, 13,649 (92.9%) were annotated using one or more databases (S5–S7 Tables), including InterProScan (*n* = 13,220), eggNOG (12,273) and Kyoto Encyclopedia of Genes and Genomes (KEGG; 9768).

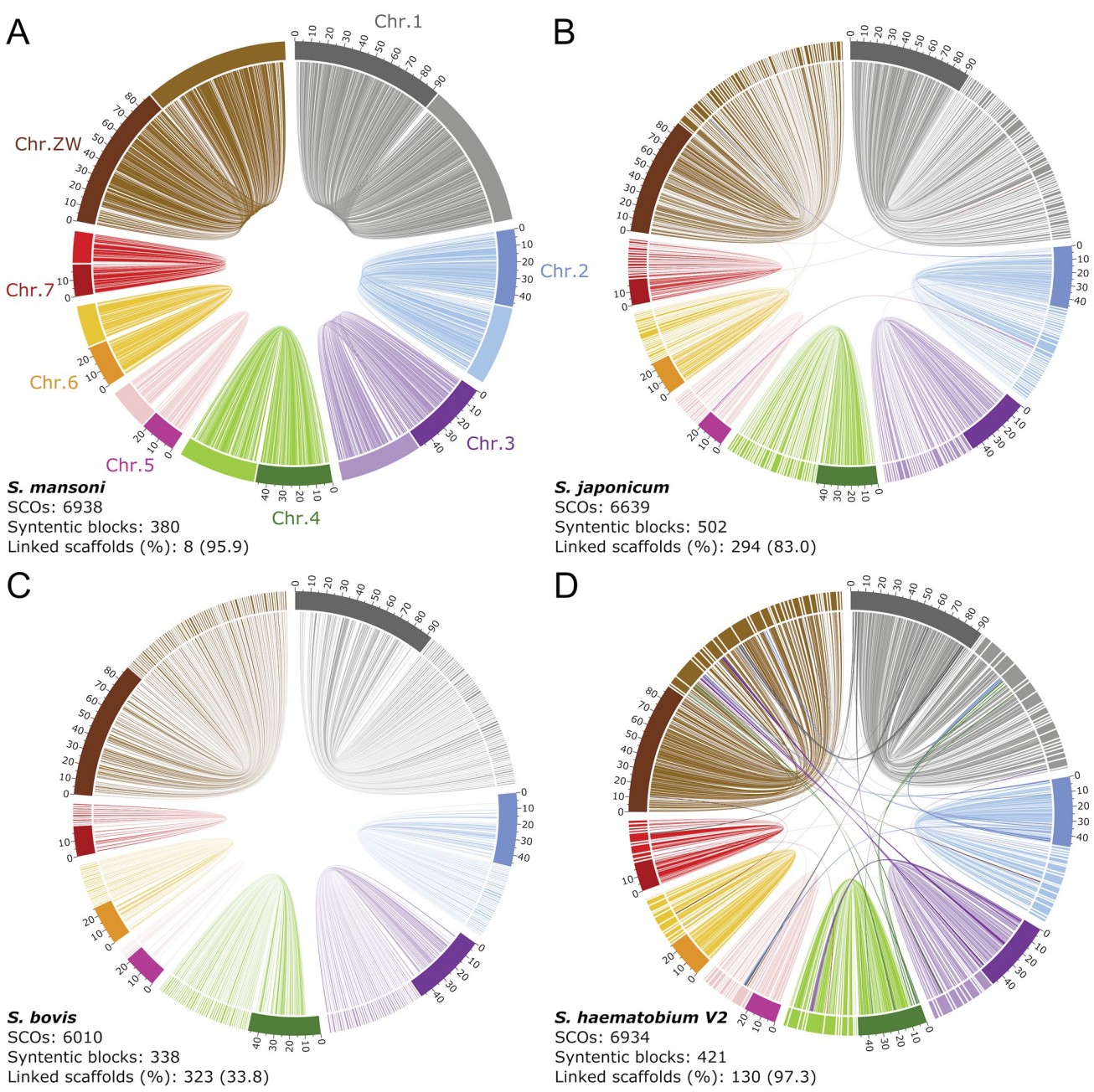

**Fig 1. Synteny and contiguity of the *Schistosoma haematobium* reference genome.** Comparisons are shown with genomes of **A**, *S. mansoni*, **B**, *S. japonicum*, **C**, *S. bovis* and **D**, the published draft genome of *S. haematobium* (Shae.V2). The eight chromosomes are represented as bars in a circular fashion, are distinctly-coloured in a dark shade and named according to the *S. mansoni* chromosomes. Syntenic blocks containing five or more single-copy orthologs (SCOs) between *S. haematobium* and the respective other species are shown as 'links' and are coloured, in a lighter shade, based on the link that spans the largest portion of the linked reference scaffold/chromosome. The number of SCOs, syntenic blocks, and linked scaffolds, as well as the percentage of the genome assembly that they represent are shown for each panel.

**Table 1. Key metrics of the *Schistosoma haematobium* Shae.V3 assembly and comparison with assemblies for other key schistosome species.**

| Metric | *S. haematobium* V3 | *S. haematobium* V2 | *S. mansoni*[b] | *S. bovis*[b] | *S. japonicum*[b] |
|---|---|---|---|---|---|
| N50 | 48,328,128 | 4,779,868 | 50,458,499 | 202,989 | 1,093,989 |
| L50 | 3 | 26 | 3 | 498 | 94 |
| N90 | 22,148,653 | 1,076,958 | 24,989,083 | 30,057 | 238,898 |
| L90 | 7 | 88 | 7 | 2299 | 348 |
| Longest scaffold | 93,306,550 | 14,276,808 | 88,881,357 | 1,115,616 | 6,264,197 |
| Shortest scaffold | 2000 | 518 | 1307 | 2009 | 1019 |
| Number of scaffolds | 163 | 666 | 320 | 4774 | 1789 |
| Genome size | 400,271,889 | 371,394,055 | 409,579,008 | 373,478,075 | 369,900,518 |
| Number of Ns | 23,062 (0.01%) | 951,002 (0.26%) | 9,332,694 (2.28%) | 12,677,721 (3.39%) | 26,673 (0.01%) |
| Number of gaps | 45 | 3128 | 282 | 16,814 | 319 |
| Repeat content | 54.3795 | 53.39 | 49.23 | 50.9114 | 46.87 |
| GC content | 35.2 | 34.4 | 34.7 | 33.2 | 33.8 |
| Complete BUSCOs[a] | 211 (82.7%) | 195 (76.5%) | 216 (84.7%) | 203 (79.6%) | 201 (78.8%) |
| Complete and single-copy BUSCOs | 208 (81.6%) | 193 (75.7%) | 211 (82.7%) | 198 (77.6%) | 200 (78.4%) |
| Complete and duplicated BUSCOs | 3 (1.2%) | 2 (0.8%) | 5 (2.0%) | 5 (2.0%) | 1 (0.4%) |
| Fragmented BUSCOs | 22 (8.6%) | 32 (12.5%) | 13 (5.1%) | 26 (10.2%) | 20 (7.8%) |
| Missing BUSCOs | 22 (8.6%) | 28 (11.0%) | 26 (10.2%) | 26 (10.2%) | 34 (13.3%) |

[a] Number of Benchmarking Universal Single-Copy Orthologs (BUSCOs) identified (genome mode), and percentage of the 255 genes within the Eukaryota data set.

[b] NCBI accession numbers: PRJEA36577, PRJNA520774 and PRJNA451066. Data sets were obtained from WormBase Parasite (release WBPS15).

**Table 2. Features of the gene and protein sets for *S. haematobium* V3, V2 and other key schistosome species**

| Feature | *S. haematobium* V3 | *S. haematobium* V2 | *S. mansoni*[c] | *S. bovis*[c] | *S. japonicum*[c] |
|---|---|---|---|---|---|
| Number of genes/mRNA | 9431/14,700 | 9314/9314 | 10,172/14,499 | 11,576/11,576 | 10,089/16,936 |
| Gene length[a] | 23,252 ± 25,748 | 18,333 ± 20,681 | 21,682 ± 24,112 | 12,618 ± 16,045 | 18,366 ± 21,336 |
| mRNA length | 3892 ± 3651 | 2195 ± 1978 | 2794 ± 2266 | 1458 ± 1501 | 2578 ± 2068 |
| Coding domain length | 1600 ± 1659 | 2004 ± 1881 | 1775 ± 1895 | 1458 ± 1501 | 1537 ± 1498 |
| Exon length | 487 ± 1118 | 263 ± 343 | 320 ± 468 | 259 ± 314 | 333 ± 540 |
| Protein length | 532 ± 553 | 666 ± 625 | 591 ± 632 | 485 ± 500 | 512 ± 499 |
| Number of 5' UTRs | 12,563 | 3097 | 14,157 | n/a[d] | 12,421 |
| Number of 3' UTRs | 12,888 | 2935 | 14,171 | n/a | 12,503 |
| Complete BUSCOs[b] | 736 (77.1%) | 639 (67.0%) | 752 (78.8%) | 577 (60.5%) | 688 (72.1%) |
| Complete and single-copy BUSCOs | 582 (61.0%) | 628 (65.8%) | 607 (63.6%) | 548 (57.4%) | 386 (40.5%) |
| Complete and duplicated BUSCOs | 154 (16.1%) | 11 (1.2%) | 145 (15.2%) | 29 (3.0%) | 302 (31.7%) |
| Fragmented BUSCOs | 26 (2.7%) | 53 (5.6%) | 24 (2.5%) | 114 (11.9%) | 43 (4.5%) |
| Missing BUSCOs | 192 (20.1%) | 262 (27.5%) | 178 (18.7%) | 263 (27.6%) | 223 (23.4%) |

[a] Lengths presented as mean ± standard deviation.

[b] Number of Benchmarking Universal Single-Copy Orthologs (BUSCOs) identified (protein mode), and percentage of the 954 genes for the Metazoa data set.

[c] NCBI accession numbers: PRJEA36577, PRJNA520774 and PRJNA451066. Data sets were obtained from WormBase Parasite (release WBPS15).

[d] not available.

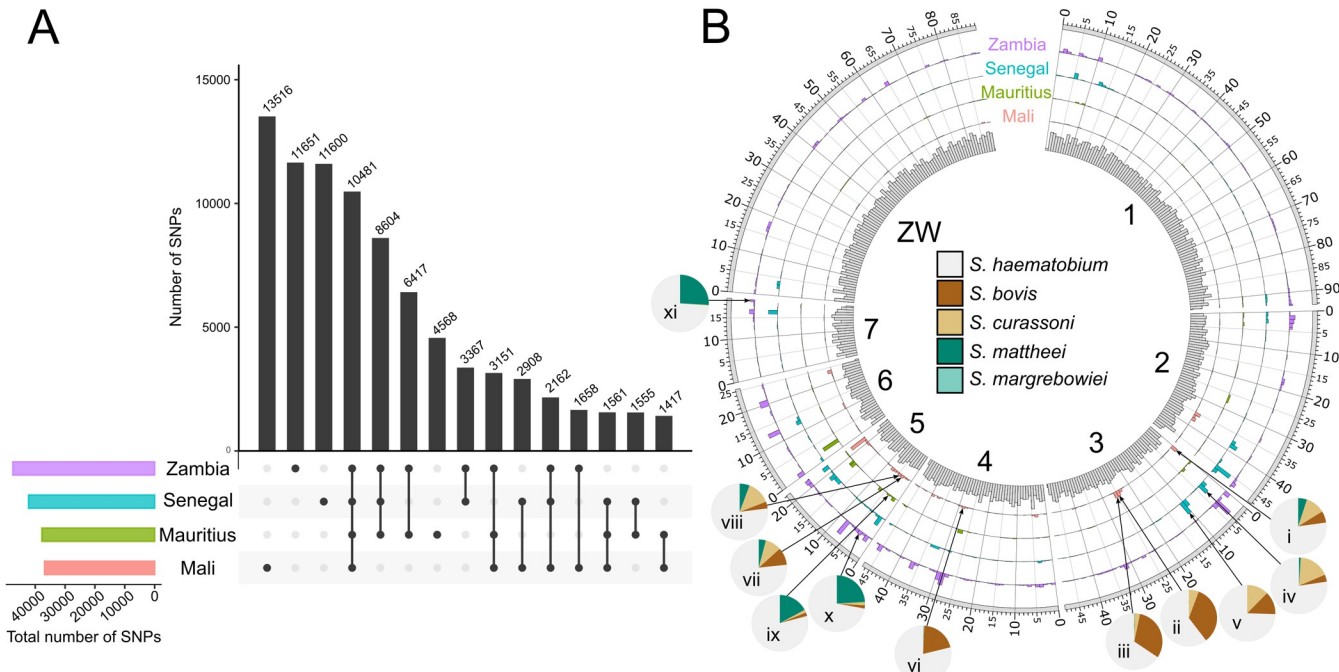

**Fig 2. Analysis of single nucleotide polymorphisms (SNPs) of four individual male *Schistosoma haematobium* worms from distinct geographic location.**
**A** Intersections of unique or shared, fixed SNPs within the predicted coding regions for isolates from Zambia, Senegal, Mauritius or Mali. Total numbers of SNPs within individual samples are indicated by distinctly-coloured bars (bottom left). **B** For all samples, density and localisation of SNPs in the *S. haematobium* reference genome are shown as histograms in the same colour. Gene densities are shown in a histogram on the innermost track, divided into 1Mb sections along each chromosome. For each sample, SNP-rich regions of which > 20% resembled a genomic reference other than *S. haematobium* are labelled (i-xi), and the distribution of matches against the genome of other schistosome species is displayed as a pie chart.

## Molecular variation among individual *S. haematobium* worms from distinct geographic locations

Using genome Shae.V3 representing the Egyptian strain of *S. haematobium* [27], we assessed the nature and extent of genetic variation between individual male *S. haematobium* from distinct geographic locations (Fig 2 and Table 3 and S1–S4 Datasets). Compared with this reference strain, we identified 1.4 to 2.0 million SNPs, a marked proportion (29.5–54.5%) of which represented fixed (i.e. unequivocally homozygous) SNPs (Table 3). Of all fixed SNPs, ~ 6% were within protein-coding regions of Shae.V3, with a notable percentage (12.1 to 36.7%) being uniquely present in individual samples (Table 3 and Fig 2A). Taken together, fixed SNPs from all four samples were found in 9129 (96.8%) of all protein-coding genes, and between 9783 (Mali-sample) and 13,564 (Zambia-sample) SNPs were inferred to have a moderate or high impact on the encoded protein (including a loss of a start codon, gain of a stop codon, or a non-synonymous alteration).

**Table 3. Summary of the single nucleotide polymorphisms (SNPs) predicted in four representative *Schistosoma haematobium* males from Zambia, Senegal, Mauritius or Mali.**

| Geographic location | Total SNPs | Fixed SNPs (GN = 1/1) | Fixed SNPs in protein-coding regions | Unique, fixed SNPs in protein-coding regions |
|---|---|---|---|---|
| Zambia | 1,415,223 | 771,957 | 47,491 | 11,651 |
| Senegal | 1,617,886 | 696,405 | 42,238 | 11,600 |
| Mauritius | 1,539,711 | 603,944 | 37,754 | 4568 |
| Mali | 2,081,064 | 613,253 | 36,854 | 13,516 |

Across the genome, SNP density was low and did not correlate with gene density; 67.6% to 80% of all SNPs per sample (individual worm) were concentrated in SNP-dense regions, collectively representing ~ 20% of the genome for each sample. Eleven of these SNP-dense regions (designated i–xi in Figs 2B and S1) contained substantial portions (> 20%) that were most similar (at the nucleotide level) to genomes of *Schistosoma* species other than *S. haematobium*. Between one and five of these regions were located on chromosomes 3, 4, 5 or 7; their location differed among samples from different geographic origins (Figs 2B and S1), with the exception of one region located on chromosome 5 (3–4 Mb) that was detected in three of the four samples (from Zambia, Mauritius and Mali). Of all samples, three SNP-dense regions identified in the Mali-sample (ii, iii and vi) and two identified in the Zambia-sample (x and xi) showed the greatest resemblance to those in the genomes of species other than *S. haematobium* but within the *S. haematobium* group; 20.6–33.9% of the SNP-dense regions ii, iii and vi matched those in *S. bovis* and 24.1–25.8% of regions x and xi matched those in *S. matthei*. For regions iv and v (Senegal-sample), as well as regions i and viii (Mali-sample), significant portions (11.1–17.6%) matched those in *S. curassoni*.

## Variations in the transcriptome among developmental stages and sexes of *S. haematobium*

We explored variation in the transcriptional profiles of protein-coding genes among seven key stages/sexes: eggs from urine; eggs from hamster tissues; sporocysts; cercariae; schistosomules; and adult male and female worms. We showed that 69% to 86.8% of all protein-coding genes were transcribed in each of these stages/sexes, with varying numbers of transcribed isoforms overall (54.4–81.2%), and per gene (1.2–1.4) (Tables 4 and S4). For each stage, a small percentage of genes (top 1%) showed substantially higher transcription (median TPM: 1391–2435) than all other genes (median TPM: 5.74–26.3) (Table 4). The functional annotation for these genes (S8 Table) mostly varied among stages/sexes, although some protein families (representing RNA transport and ribosomal proteins, for instance) were represented by the top 1% transcripts in more than one stage/sex (Tables 4 and S8).

**Table 4. Summary of transcription levels across seven key developmental stages of *Schistosoma haematobium*.**

| Developmental stage | Number of transcribed genes[a] (%) | Number of transcribed isoforms (%) | Average (mean) number of transcribed isoforms per gene | Median TPM[b] | Median TPM of top 1% transcribed isoforms | Key protein/pathway functions for top 1% transcribed isoforms |
|---|---|---|---|---|---|---|
| Egg (from urine) | 8153 (86.4) | 11,106 (75.6) | 1.4 | 15.0 | 1790 | Translation; RNA transport; ribosomal proteins |
| Egg (from hamster) | 7446 (79.0) | 9584 (65.2) | 1.3 | 9.95 | 1592 | Ubiquitin; protein folding, sorting and degradation; RNA transport; ribosomal proteins |
| Sporocyst | 6506 (69.0) | 7990 (54.4) | 1.2 | 5.95 | 1894 | Cellular nucleic acid-binding protein; RNA transport; ribosomal proteins |
| Cercaria | 7202 (76.4) | 9280 (63.1) | 1.3 | 5.74 | 1607 | Calmodulin; cytochrome *c* |
| Schistosomule | 7696 (81.6) | 10,657 (72.5) | 1.4 | 26.3 | 1391 | Peptidyl-prolyl isomerase; 14-3-3 protein beta; RNA transport, ribosomal proteins |
| Adult male | 8182 (86.8) | 11,935 (81.2) | 1.5 | 9.81 | 1702 | Glutathione S-transferase; peptidases/proteases |
| Adult female | 8112 (86.0) | 11,714 (79.7) | 1.4 | 7.79 | 2435 | Peptidases/proteases |

[a] TPM > 0.5

[b] transcripts per million

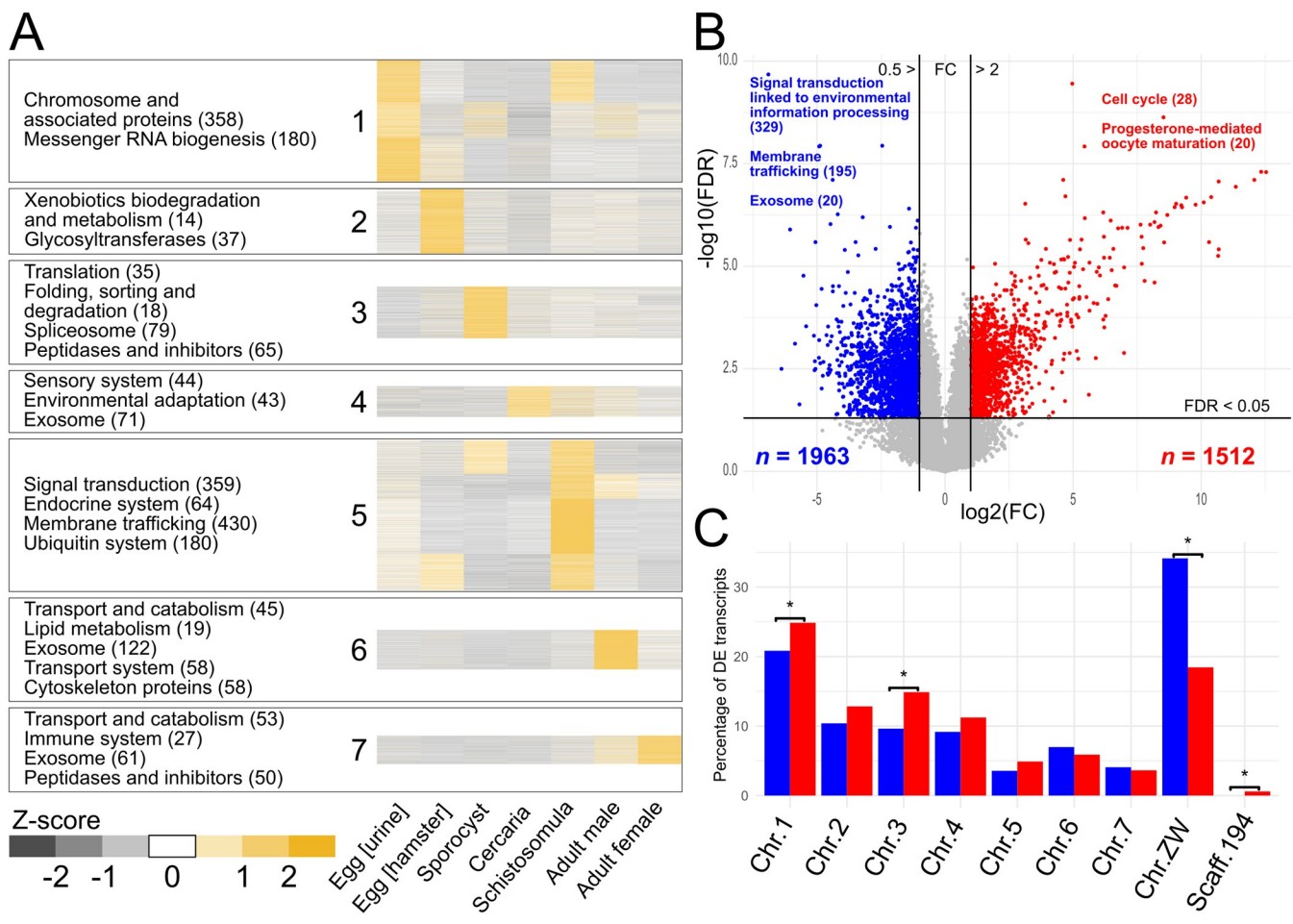

**Fig 3. Analysis of transcription for key developmental stages of *Schistosoma haematobium*. A** Transcription profiles of transcript isoforms across seven developmental stages/sexes, clustered (Ward; *k* = 7) by similarity of Z-score-normalised TPM (transcripts per million) values. Key, enriched (q < 0.05) pathways and/or protein functions are shown to the left of each cluster. Numbers of molecules in round parentheses. **B** Pairwise comparison of differential (DE; fold change (FC) > 2, false discovery rate (FDR) < 0.05) transcription between male (blue) and female (red) samples, displayed as a 'volcano' plot. Key pathways and/or protein functions enriched in DE subsets are highlighted. **c** Percentage of DE transcripts encoded on each chromosome/scaffold for males (blue) and females (red), respectively; chromosomes/scaffolds enriched (q < 0.05) for male or female DE genes are marked with an asterisk.

## Genes with similar transcription profiles and functions among developmental stages

We hypothesised that transcription profiles that correlated among different developmental stages would link to common pathways or signalling networks. We established seven distinct clusters, each containing 817 to 4301 transcripts with similar transcription profiles (Fig 3A and S4 Table). Each of the seven clusters contained transcripts predominantly transcribed in one of the seven key developmental stages. For clusters 1 and 5, marked co-transcription was seen among two to three different stages. By contrast, 22, 3 and 3 transcripts were uniquely transcribed in the sporocyst, cercaria and schistosomula stages, respectively (clusters 3–5, S4 Table). In the sporocyst stage, unique transcripts encoded peptidases/proteases, CAP domain-containing proteins (including "venom allergen-like" or SmVAL-like proteins) and a heat shock protein-associated CDC37 homolog (MS3_00009347.2). CAP protein-encoding (MS3_00004475.1) and sodium channel-encoding (MS3_00007597.2) transcripts were unique

to the cercarial stage, and a transcript exclusive to the schistosomula stage encoded a "sperm-tail PG-rich repeat" protein (MS3_00007199.1).

Overall, all seven clusters showed significant enrichment for protein families (*n* = 35) and/or pathways (47) (Fig 3A and S9 Table), including those linked to transport and catabolism in both adult stages (clusters 6 and 7), as well as molecules related to the exosome in the cercaria stage and both adult sexes (clusters 4, 6 and 7). Additionally, peptidases and peptidase inhibitors were enriched in clusters 3 (sporocyst) and 7 (adult female).

## Distinct isoform usage in different developmental stages

Next, we investigated genes that encoded distinct isoforms in multiple transcription-clusters, reflecting variation in isoform usage for different developmental stages/sexes. We identified 2648 of such genes with distinct isoforms present into two (*n* = 2102) to six (1) clusters (S4 Table). We hypothesised that this isoform switching is driven by alternative splicing and is either facilitated by genes encoding small exons (microexons of ≤ 54 nt) or many exons. Although there was no overall correlation between the number of isoforms in distinct clusters and the number of microexons (Pearson's R = 0.23) or exons (R = 0.3) per gene, we did find evidence of genes encoding microexons and multiple isoforms in distinct clusters. For example, of the genes containing inferred microexons (*n* = 2996), two genes encoded isoforms that were present in five of the seven clusters, and were comprised of 7–10 exons and 0–2 microexons. These isoforms were assigned to clusters 2–6 (MS3_00008061) and 3–7 (MS3_00004678), and encoded a MYND-type zinc finger domain-containing protein (IPR002893) and a small GTPase (IPR001806), respectively. We provided evidence for the differential usage of two isoforms transcribed predominantly in males (cluster 6) or females (cluster 7), respectively, by mapping transcripts assembled from mixed-sex, long-reads to the genomic region encoding MS3_00004678 (chromosome ZW, positions 87,941,969 to 87,954,222; Fig 4).

## Sex-linked transcription

A comparison of transcription levels in the male and female adult stages of *S. haematobium* showed that 1512 transcripts were significantly upregulated in female compared with male

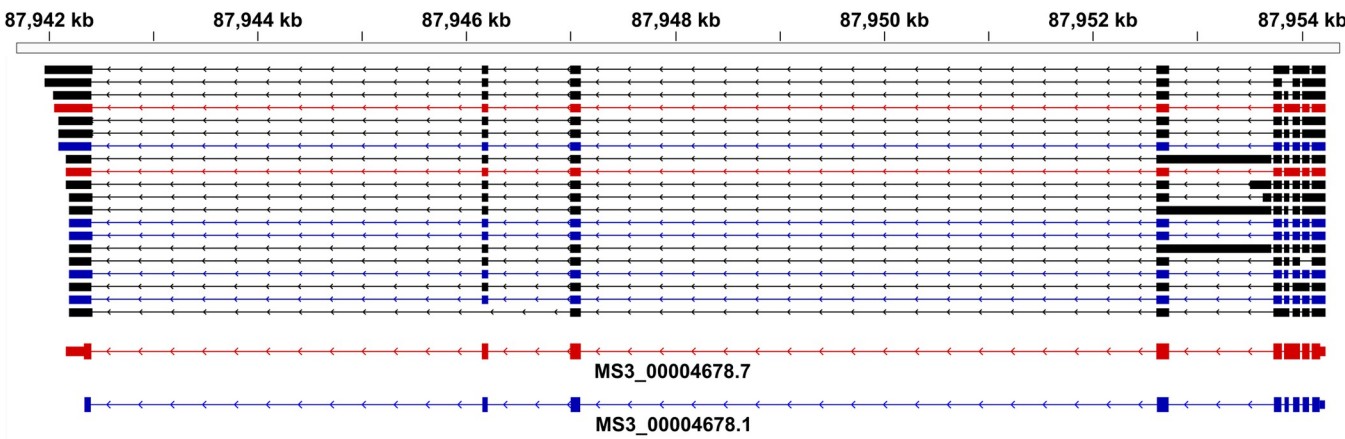

**Fig 4. Long-read, full-length transcripts supporting differential isoform usage in male and female *Schistosoma haematobium*.** The gene model MS3_00004678 encodes a small GTPase on chromosome ZW. Exons are depicted as blocks and introns as arrowed lines, indicating the coding strand. Reference transcripts are shown at the bottom in red (female; MS3_00004678.7, transcription cluster 7) and blue (male; MS3_00004678.1; transcription cluster 6) with narrow blocks at the end of the gene models representing untranslated regions (UTRs). Full-length, long-read transcripts that matched the intron-exon structure of the isoforms inferred to be transcribed in the male and female adult stage, respectively, are coloured accordingly. Transcripts that support distinct, alternative exon-intron boundaries are shown in black.

worms (Fig 3B and S10 Table), and that the genes encoding these transcripts were over-represented (Fisher's exact test, adjusted p-value < 0.05) on chromosomes 1 and 3 and on the largest, "unplaced" scaffold (no. 194) of Shae.V3 (Fig 3C). None of the 10 genes encoded in this scaffold were upregulated in males, and all had very low transcription levels (mean TPM of ≤ 0.57) in males.

In *Schistosoma*, sex is determined by a ZW chromosomal system, whereby maleness is conferred by a ZZ composition and the absence of the female-specific W chromosome. Unlike in most XY systems, where double-dosage of X transcripts is prevented by transcriptional suppression in XX individuals, schistosomes have a limited suppression of Z chromosomes [28]. Consistent with a ZW chromosomal system, genes located on the *S. haematobium* ZW chromosome encoded markedly more of the 1963 sexually-upregulated transcripts, with 670 (34.1%) upregulated in males and 279 (18.5%) in females (Fig 3B and 3C and S10 and S11 Tables).

Of the transcripts upregulated in females, there was significant enrichment of those encoding proteins linked to progesterone-mediated oocyte maturation, cell cycle and ribosome, and proteins involved in DNA replication and chromosome-related functions (S12 Table). Proteins encoded by transcripts upregulated in male worms were significantly enriched for roles in 50 different pathways, including those involved in signal transduction associated with environmental information processing (329 proteins), and endocrine (251), nervous (148) and digestive (106) systems linked to 5 to 11 pathways. Enriched protein families included those related to the cytoskeleton ($n$ = 99), transport (membrane trafficking: $n$ = 195; exosome: 184; transport system: 95) and signalling (ion channels: $n$ = 45; G protein-coupled receptors: 31) (S12 Table).

## Distinctive transcription in eggs, depending on host origin

We hypothesised that the transcription in *S. haematobium* eggs derived from human urine would differ from eggs isolated from hamster livers. A comparison revealed 1143 transcripts that were unique to eggs from urine, including a hepatotoxic ribonuclease *omega*-1 (UniProt identifier: Q2Y2H5) homolog (69% amino acid similarity (BLAST); MS3_00010006.1; TPM = 11.0–111.9). To investigate whether this homolog was structurally similar to the *S. mansoni omega*-1 protein, we predicted the structures of all seven *S. haematobium omega-1* homologs using AlphaFold [29]. The alignment of these predicted structures using TM-align [30] showed that six of these homologs (including MS3_00010006.1) aligned well (RMSD: 1.15–2.2Å; TM-score: 0.69–0.85) with 74.7–90.2% of the *S. mansoni omega-1* structure, despite limited overall sequence identity (33.5%-50.6%, based on structural alignment) (S13 Table).

Additional 4965 transcripts had substantially higher TPMs (> 112) in eggs from human urine, including an "interleukin (IL)-4-inducing principle of *S. mansoni* eggs" (M-IPSE/*alpha-1*; UniProt identifier: Q869D4) homolog (H-IPSE; MS3_00010265.1). The proteins encoded by 6108 transcripts linked predominantly to spliceosome ($n$ = 89 proteins), membrane trafficking (221), as well as transferase (537) and hydrolase (443) activities. Enriched were protein families that related to chromosomes ($n$ = 358) and mRNA biogenesis ($n$ = 180). By contrast, a *kappa-5* (UniProt identifier: Q2KMJ3) homolog (MS3_00010619.1) had a higher level of transcription (TPMs > 112) in eggs from hamster liver than from human urine.

## Discussion

The assembly of the chromosome-contiguous reference genome (Shae.V3) for a well-defined Egyptian strain [27] of *S. haematobium* has underpinned an exploration of molecular variation within *S. haematobium* at key stages of development from different hosts and from multiple

geographic localities in Africa, with important implications for investigating natural schistosome populations as well as urogenital schistosomiasis and associated bladder cancer in humans.

The substantial genetic variation observed among four *S. haematobium* samples from four disparate locations in Africa (Zambia, Senegal, Mauritius and Mali) was associated with unique genomic 'signatures' matching species other than *S. haematobium*. This finding supports the proposal that schistosome species within the *S. haematobium*-group form a complex genetic landscape, resulting from genomic admixture and introgression upon hybridisation [21,31]. The presence of such hybridisation/introgression events raises the importance of exploring natural populations of members of this group and establishing their biological traits in relation to host affiliations/range, pathogenicity, susceptibility to praziquantel and, particularly, carcinogenicity. In this context, the fragmented nature of the existing assemblies for some members of the *S. haematobium*-group and the lack of draft or reference genomes for *S. guineensis*, *S. intercalatum* and *S. leiperi* represents a hurdle to more detailed explorations of the extent and size of such introgression events. Clearly, future genome sequencing efforts should place emphasis on creating reference genomes for all other members of the *S. haematobium* group, to complement the *S. haematobium* reference genome (Shae.V3).

The present genome, comprehensive transcriptomic profiling and long-read evidence allowed us to refine gene models and annotations, discover 'new' genes ($n$ = 45) and define UTRs, which will enable further molecular explorations of *S. haematobium*. Variation in the transcription profiles of genes likely relate to molecular alterations during developmental, infection and/or disease processes. For instance, genes exclusively transcribed in the sporocyst, cercaria and schistosomule stages encoding peptidases/proteases (including leishmanolysins, metalloendopeptidases and trypsins) or SCP/TAPS superfamily members (e.g. venom-allergen like proteins, VALs [32]) likely play roles in egress, invasion, digestive processes and/or immune evasion in the molluscan or vertebrate hosts [33–35]. Sex-specific molecules identified likely associate with roles in development and/or reproduction in the female, and signalling, transport and catabolism in the male [25,36–38]. It is noteworthy that many genes on the Z chromosome are upregulated in ZZ males, consistent with a lack of widespread transcriptional dosage compensation of the Z chromosome [28]. The lack of transcription in 'male' genes encoded on the largest "unplaced" scaffold (no. 194) of Shae.V3 suggests that this scaffold represents a female-specific portion of the W chromosome. However, the complete W-specific region (WSR) is likely much larger based on evidence for *S. mansoni*, whose highly-repetitive WSR is estimated at 18–46 Mb [39]. Future work is warranted to fully resolve the sex chromosomes of *S. haematobium* using long-read data from individual worms (females and males) as a foundation for detailed explorations of sex-determining genes and sex- and developmentally-regulated gene expression.

We propose that variation in transcription levels between eggs from hamster liver and those from human urine relate to differences in host-parasite relationship and to the ability of eggs to induce immunopathological changes and disease (which is pronounced in humans, but not in the hamster), including the presence of *S. mansoni* homologs of IPSE/*alpha-1* [40,41], *kappa-5* [42] and ribonuclease *omega-1* [43]. Intriguingly, as *omega-1* was not detected previously in proteomic investigations of egg-secreted antigens (ESAs) of *S. haematobium* [44,45], or known to induce a humoral antibody response in people and not detected in the urine from *S. haematobium*-infected people [44], this ESA was considered as *S. mansoni*-specific [40]. However, to some extent, structural modelling supports the presence of those molecules in *S. haematobium* eggs from human urine. Whether the transcripts of these homologs are specifically transcribed in the eggs from urine from infected human patients and encode immunogens that involved in the egg-directed immune responses in the human host warrants investigation. In this context, the enriched transferase-encoding transcripts in urine-derived *S.*

*haematobium* eggs might relate to roles in glycosylation of immunomodulatory glycoproteins such as *omega-1* and *kappa-5*, likely required for protein function, as described for *S. mansoni* [46,47]. The findings from this study lay a critical foundation for investigation of ESAs in *S. haematobium* and can complement efforts to understand the pathogenesis of urogenital schistosomiasis [48–50].

The chromosome-level genome assembly for an Egyptian strain of *S. haematobium* adds important resources to the schistosome '-omics' reference toolkit. For example, this genome should accelerate large-scale population investigations and provide a unique opportunity to study the implications of genomic admixture, including its effect on biological and/or disease traits, morbidity and/or the effectiveness of control programs [51,52], including mass drug administration (MDA) [53]. The present resource should also enable future functional genomics investigations of *S. haematobium* [54–56] and facilitate investigations of the fundamental pathobiology of this important parasite using an integrative proteomic, glycomic and lipidomic approach. Insights into these areas could significantly assist in ongoing control and elimination efforts of urogenital schistosomiasis. We expect that the long-read sequencing technologies used herein will facilitate future investigation of schistosome chromosomes and transcriptomes, particularly differential isoform transcription and alternative splicing in sex determination, development and reproduction.

## Methods

### Ethics statement

Approval to maintain the life cycle of *S. haematobium* using *Mesocricetus auratus* (hamster; mammalian definitive host) and *Bulinus truncatus* as the snail intermediate host at the Biomedical Research Institute (BRI), Rockville, Maryland, USA was obtained from the NIH Office of Laboratory Animal Welfare [OLAW]: D16-00046 (A3080-01). Ethics approval for the collection of blood fluke parasite materials for the Schistosomiasis Collection at the Natural History Museum (SCAN) was obtained from the Home Office, project license number 70/4687 [14]. Approval to collect urine from schoolchildren was obtained from the administrative authorities, school inspectors, directors and teachers. The objectives of the study were explained to schoolchildren and their parents or guardians, and to participants from whom written informed consent was obtained. The study was also approved by the National Ethics Committee (Nr 2016/11/833/CE/CNERSH/SP) and the Ministries of Health and Education of Cameroon, and from the Liverpool School of Tropical Medicine Research Ethics Committee (M1516-18 and M1516-06).

### Parasite material

Different developmental stages of *S. haematobium* were obtained from experimental and natural hosts and distinct geographical regions. Adult, egg and schistosomule stages of *S. haematobium* originating from Egypt) [27] and maintained routinely in *M. auratus* (hamster; mammalian definitive host) using *B. truncatus* as the snail intermediate host at the BRI, Rockville, Maryland, USA. Hamsters exposed to 1,000 cercariae in pond water (200 ml) were euthanised after 90 days of infection. Paired *S. haematobium* adults were perfused from the mesenteric/intestinal vessels with physiological saline (37˚C) using an established method [25]. Schistosomules were prepared by mechanical transformation [57] of ~10,500 cercariae shed from infected *B. truncatus*, followed by culture for 24 h [58]. All of these developmental stages were prepared and stored at -80˚C or -196˚C.

Single adult males of *S. haematobium* from four disparate geographic locations in Africa (Zambia, Senegal, Mauritius and Mali) were obtained *via* SCAN [59]. Adult worms were

perfused at 90 days from *M. auratus* infected in the laboratory with from individual *Bulinus wrighti* snails infected with miracidia from eggs from urine samples from individual patients (*n* = 3), or from hamsters infected with cercariae from naturally infected snails (*B. truncatus*) (*n* = 1) (S14 Table). These worms were frozen in liquid nitrogen until use.

Eggs were collected from the urine from ~6 to 10 year-old children attending schools near Loum, Cameroon, with approval from the administrative authorities, school inspectors, directors and teachers. Individual eggs were isolated microscopically and stored in RNA*later* at 4°C (Thermo Fisher Scientific, Waltham, MA, USA).

## DNA sequencing and genome assembly

The *S. haematobium* reference genome (designated Shae.V3 – representing the Egyptian reference strain [27], maintained at BRI) was assembled from data produced by Oxford Nanopore long-read and Hi-C sequencing and from previous short-read data sets produced using Illumina or Dovetail technology [25,26] using the following approach:

First, long-read data (SRA accession numbers SRR15400746 and SRR15400747; *via* Oxford Nanopore technology [60]) were used for initial contig assembly employing the program Canu v.1.9 [61], setting a genome size estimate of 400 Mb. Errors in these data were corrected using medaka_consensus in the Medaka package v.1.0.3 (https://github.com/nanoporetech/medaka). Redundancy was removed using purge_haplotigs v.1.1.1 [62] and using depths of 8, 35 and 100 reads (low, medium and high, respectively). Contigs were first scaffolded using available short-read and mate-pair library data [25] using Platanus-allee v.2.0.2 [63], using a minimum of 15 links to join contigs into contiguous scaffolds. Further scaffolding was done using long-range, paired-read data ('Dovetail') using the HiRise pipeline v.2.0.5 [64], as described earlier [26]. Then, scaffolds were polished using available short-read and mate-pair library data employing pilon v.1.23 [65].

Next, *in situ* Hi-C sequencing was performed as described previously [66]. High molecular weight DNA from 100 *S. haematobium* adults was restriction-digested with equal concentrations of *Cvi*AII and *Mse*I (New England Biolabs); the library was constructed and then sequenced using the NextSeq550 platform (Illumina, San Diego, CA, USA). Scaffolds were combined with the *in situ* Hi-C data using Juicer v.1.6 [67], 3D-DNA v.180922 [68] and Juicebox Assembly Tools v.1.9.8 [69] to scaffold, inspect and manually curate results to achieve chromosome-length scaffolds. The sequence data are available *via* the DNA Zoo SRA repository (PRJNA512907); interactive Hi-C contact maps before and after the Hi-C-guided assembly are available on the DNA Zoo website (https://www.dnazoo.org/assemblies/Schistosoma_haematobium). Gaps in scaffolds were closed using long-reads that had been error-corrected using the *-correct* and *-trim* steps within the program Canu employing the program TGS-Gap-Closer v.1.0.3 (https://github.com/BGI-Qingdao/TGS-GapCloser). The gap-closed scaffolds were then polished employing published data sets (produced from 500-bp and 800-bp libraries) [25] and the error-corrected long-reads using the software HyPo v.1.0.3 [70].

Repeats in the final, gap-closed and polished assembly were identified and masked using RepeatMasker v.4.1 (http://www.repeatmasker.org) employing the DFAM v.3.1 library and a published *S. haematobium* repeat [25].

## Synteny analysis

Genome-wide synteny between the repeat-masked Shae.V3 genome and the repeat-masked scaffolds or chromosomes of other schistosome species was assessed by linking single-copy orthologs (SCOs) (for each species-pair). Coordinates of SCOs were used as links between scaffolds and were bundled using bundlelinks in circos tools v.0.23 [71], setting the minimum

bundle size at 10,000 nt, with ≥ 5 SCOs per bundle, and allowing the gap between members of the same bundle to be at most 100,000 nt. Scaffolds were ordered and displayed using circos v.0.69–8 [71].

## RNA sequencing and data sets

Total RNA samples were isolated from (i) adult worms (50 worm pairs; three biological replicates), (ii) individual male and female worms separated from pairs (six biological replicates for each sex), (iii) cercariae and (iv) mechanically-transformed schistosomules of *S. haematobium* using the TriPure Isolation Reagent (Sigma Aldrich, St. Louis, MO, USA). Each RNA sample was treated with DNase (TURBO DNA-free™ kit, Thermo Fisher Scientific, Waltham, MA, USA) and messenger RNA (mRNA) was purified (Dynabeads mRNA purification kit, Thermo Fisher Scientific, Waltham, MA, USA). The size, integrity (i.e. RNA integrity number, RIN) and concentration of RNA were estimated using a 4200 TapeStation System RNA ScreenTape Assay (Agilent Technologies, Waldbronn, Germany) and a Qubit 3.0 Flourometer RNA High Sensitivity Assay (Life Technologies, Carlsbad, CA, USA). TruSeq Stranded mRNA (Illumina, San Diego, CA, USA) short-read libraries (150 bp, paired-end) were prepared from from individual mRNA samples, according to the manufacturers' instructions and sequenced on an Illumina NextSeq 500 instrument.

Total RNA samples were also prepared from *S. haematobium* eggs (~ 500 to 1000 each), isolated from urine samples from three different individuals, using the TRIzol Plus RNA purification kit (Thermo Fisher Scientific, Waltham, MA, USA) and non-stranded, paired-end libraries (145 bp) were constructed (TruSeq Non-Stranded Kit, Illumina, San Diego, CA, USA) and sequenced on an Illumina HiSeq 4000 platform at BGI International (Shenzhen, China).

All short-read data produced here were filtered for quality and adapters removed using the program fastp v.0.20.1 [72]. Then, reads representing technical artefacts (including PCR duplicates) or contamination were removed by mapping all quality-filtered and trimmed reads to published genome scaffolds [26] using HISAT2 v.2.1.0 [73] with the options-fr for upstream/downstream mate orientations for Illumina paired-end sequencing and-dta ("downstream transcriptome analysis"). Mapped reads were then retained by filtering sam files using the -F4 flag in samtools v.1.9 [74] and the remaining reads were separated into files with mapped, paired reads and mapped, unpaired reads using the options -f1 and -F1, respectively. The program centrifuge v.1.0.4 [75] was then used to confirm no contamination was present.

Publicly-available short-read data sets for (i) *S. haematobium* eggs isolated from hamster liver, pooled adult female or male worms of *S. haematobium* and pooled sporocysts produced previously [25,76] were obtained from the Short Read Archive (SRA; accession nos. SRR6655493, SRR6655495, SRR6655497 and SRR13147979).

Long-read RNA sequence data were produced from mRNA from pooled adult worms (both sexes) using Oxford Nanopore technologies (Oxford, UK). Two direct RNA-sequencing libraries using the SQK-RNA002 kit (which selects for full-length mRNAs with polyA tails), and one cDNA-PCR long-read sequencing library using the SQK-PCS109 kit were constructed. PCR-amplification (SQK-PCS109) was conducted for 14 cycles using an extension time of 3 min. All libraries were sequenced using a MinION device for 48–72 h using an EXP-FLP002 flow cell priming kit and three R9.4.1 flow cells (FLO-MIN106). Reads were obtained from raw fast5 files using a GPU-enabled version of the program Guppy v.3.2.4, providing the configuration file rna_r9.4.1_70bps_hac.cfg (for SQK-RNA002) or dna_r9.4.1_450bps_hac.cfg (for SQK-PCS109). Reads that did not meet the quality required (Q ≥ 7) by Guppy were removed.

## Prediction of protein-coding genes

Gene models predicted for the *S. haematobium* Shae.V2 draft genome [26] were transferred to the new genome assembly (Shae.V3) using liftOver (release 8 April 2020; [77]). First, a chain file was created using the published [26] and new genome assemblies and using the doSame-SpeciesLiftOver.pl script. Next, Shae.V2 gene models were transformed from genome feature format (GFF) to gene prediction (GP) format and transferred to the Shae.V3 genome using the liftOver chain file.

For gene prediction, quality-filtered and mapped paired-end reads from all 24 short-read libraries were combined and supplied to the programs StringTie v.2.1.4 [78] and TransDecoder v.5.5.0 [79]. Then, to infer transcripts from long-reads, long-reads were mapped to the reference genome using minimap2 v.2.17-r941[80] employing the options *-ax splice*, *-uf* and *-k14*. The program FLAIR (release Oct 2020) [81] was subsequently employed to correct splice junctions created by mapped long-reads using high-quality, mapped short-reads and to collapse mapped long-reads into transcripts using the *-stringent* option.

Gene models transferred from Shae.V2 and those inferred based on short- and long-read RNA-Seq evidence were merged using StringTie with the *-merge* option and were used as 'hints' for gene prediction using the software AUGUSTUS v.3.4.0 [82]. Next, to create a training set for AUGUSTUS, redundant, duplicate, and incomplete gene models and transcript isoforms were removed, retaining only the most highly transcribed isoform per gene and those that had a transcripts per million (TPM) value of $\geq 1$ and were covered by mapped reads across their entire length. Additionally, for each gene the isoform with highest sequence identity to a *S. mansoni* transcript sequence was also retained. Gene models that did not pass the NCBI quality checks using the program table2asn v.25.8 (https://www.ncbi.nlm.nih.gov/genbank/tbl2asn2/) were removed.

Genes were predicted using AUGUSTUS with the *- species schistosoma2* option and were subsequently refined by adding UTRs and transcript isoforms using the program PASA (docker image 8b604b34971f) [83] employing long-read transcripts as evidence. All non-redundant, complete gene models from the initial StringTie predictions and the AUGUSTUS/PASA predictions were retained as the final gene set. The completeness of the gene set was assessed using the program BUSCO v.4.0.6 [84] using the *-l metazoa_odb10* (release 10 Sept 2020) and *- update-data* options and was compared to published gene sets of *S. haematobium*, *S. mansoni*, *S. japonicum* and *S. bovis*.

## Functional annotation of inferred proteins

Protein sequences conceptually translated from predicted gene models were functionally annotated using an established approach [85]. In brief, protein sequences were assessed for conserved protein domains using InterProScan v.5.44–79.0 [86] employing default settings. Next, using the program diamond v.0.9.24.125 (E-value $\leq 10$–8), amino acid sequences were searched against the Kyoto Encyclopedia of Genes and Genomes (KEGG) database [87] to infer pathway associations, and against Swiss-Prot within UniProtKB [88] to infer homologs. Additionally, EggNOG mapper v.5.0 [89] was used to name protein sequences based on their closest match to the EggNOG database [90].

Orthologs between the inferred proteome for Shae.V3 and available proteomes of *S. mansoni* [91]; NCBI accession number PRJEA36577), *S. japonicum* [92]; NCBI accession number PRJNA520774) and *S. bovis* [31]; NCBI accession number PRJNA451066) downloaded from WormBase Parasite (release WBPS15; [93]) were determined using OrthoFinder v.2.5.2 [94].

## Analysis of genetic variation within *S. haematobium* among disparate geographic locations

High molecular weight genomic DNA was isolated from single adult males of *S. haematobium* from four distinct geographic locations (Zambia, Senegal, Mauritius and Mali) using the Chemagic STAR DNA Tissue kit (Perkin Elmer, Waltham, MA, USA). The DNA yield was estimated spectrophotometrically using the Qubit 3.0 Flourometer dsDNA HS kit (Life Technologies, Carlsbad, CA, USA), and DNA integrity was assessed by agarose-gel electrophoresis and then using a Bioanalyzer 2100 (Agilent Technologies, Waldbronn, Germany). High-quality genomic DNA was used to construct short-insert libraries (500 bp) using a TruSeq DNA library construction kit (Illumina, San Diego, CA, USA) and paired-end sequenced as 100 nt reads using the HiSeq-2500 platform (Illumina, San Diego, CA, USA).

Low-quality bases (Phred quality: < 20), adapters and reads of < 70 nt in length were removed using Trimmomatic v.0.32 [95], and sequence quality was confirmed using FastQC v.0.11.2 (http://www.bioinformatics.babraham.ac.uk/projects/fastqc/). Subsequently, high-quality reads were mapped to scaffolds of the Shae.V3 genome using Bowtie2 v.2.4.2 [96], and read alignments were stored in the BAM format. The mapped data were then used to record single nucleotide polymorphisms (SNPs) at individual positions in relation to the references using the Genome Analysis Toolkit (GATK v.4.0.8.1; [97]). In brief, base-quality scores of 'raw', aligned read data were re-calibrated twice based on predicted variants; subsequently, SNP sites were identified for each sample using the GATK HaplotypeCaller [97] and merged into one 'variant call format' (VCF) file listing all variable sites for all samples using GATK CombineGVCFs and GenotypeGVCFs. Raw SNP sites were filtered for quality using GATK VariantFiltration and following GATK best-practice guidelines. Specifically, SNP sites were selected if read mapping depth (DP) was > 10, variant confidence (QD) > 2.0, strand bias (FS) < 60.0, mapping quality (MQ) > 40.0, mapping quality (MQRankSum) > -12.5 and read position bias (ReadPosRankSum) > -8.0. VCF files for reported SNPs in each sample were annotated based on their genomic locations and predicted coding effects using snpEff v.5.0e [98] and a GFF annotation file for the reference genome. Descriptive statistics were obtained from snpEff output and using bcftools v.1.11 [74] and filtered VCF files.

The fixed SNPs (genotype call = 1/1) for each individual male of *S. haematobium* were selected and transferred onto the reference sequence using FastaAlternateReferenceMaker in GATK v.4.2.0.0. The genomic locations of fixed SNPs in coding regions were then compared within and among the four individuals, and were displayed using the UpSet v.1.4.0 package in R [99]. For each sample and each *S. haematobium* chromosome, the number of SNPs per 1Mb non-overlapping region was determined, and regions with equal or more SNPs than 80% of all 1Mb regions per chromosome (80th percentile) were selected as 'SNP-dense' locations. Each chromosome was then fragmented into 2000 nt sections and nucleotide similarity searches were undertaken using minimap2 (-x asm20 -N 5- secondary = no) and a nucleotide database of schistosome genomes, which consisted of the available genomes of key members of the *S. haematobium* group (*S. haematobium* (Shae.V3, this study), *S. bovis* (PRJNA451066), *S. curassoni* (PRJEB519), *S. mattheei* (PRJEB523) and *S. margrebowiei* (PRJEB522) as well as *S. mansoni* (PRJEA36577), *S. rodhaini* (PRJEB526) and *S. japonicum* (PRJNA520774).

The number of unique SNPs in coding regions within 2000 nucleotide regions along each chromosome were then plotted and labelled according to the species with the greatest nucleotide sequence homology match (requiring > 90% query coverage) using ggplot2 in R. SNP-dense locations for which > 20% of the 2000 nt sections (i.e. > 100 sections) matched those of a species other than *S. haematobium*, were considered to have a 'non-*S. haematobium* SNP signature'. For these regions, the number of matches against each species in the database was

represented in a pie chart. To assess the extent of false-positive species signatures in SNP-dense regions, sequence regions were also subjected to homology searches against a reference with no mutations (identical to the Shae.V3 genome sequence) and against one containing random mutations introduced at the rate of 1934 nucleotide mutations per 1 Mb of genome scaffold using msbar in the emboss package v.6.6.0.0 [100].

## Analysis of transcription

For each developmental stage, we aligned length- and quality-filtered, short-read data to the Shae.V3 genome using HISAT2, and inferred the transcription level for each transcript employing StringTie2 and the Shae.V3 gene set GFF file. Transcripts were clustered employing the Ward clustering method based on the Euclidian distance of their TPM values, that were Z-score-normalised across seven developmental stages. For stages with multiple samples, the median TPM was employed. TPM values were then ordered according to their cluster membership and displayed in a heatmap using the tidyheatmap package (https://github.com/jbengler/tidyheatmap) in R.

Differential transcription analysis for libraries derived from individual male and female worms (six biological replicates each) was conducted using Ballgown v.2.22.0 [101], employing a two-group comparison and performing library size adjustment by using the sum of the log non-zero expression measurements for each sample, up to the 75th percentile of those measurements. Transcripts with a false discovery rate of $< 0.05$ and a fold-change (FC) of $\geq 2$ were considered differentially transcribed (i.e. upregulated). Individual stages/clusters were tested for enrichment of KEGG pathways and KEGG BRITE terms (requiring a minimum BRITE protein family size of 10), using Fisher's exact test and correcting for multiple testing by calculating the q-value and applying a cut-off of $< 0.05$.

## Supporting information

**S1 Fig. Analysis of *Schistosoma haematobium* genome regions of four individual male worms from distinct geographic locations.** For isolates from Zambia, Senegal, Mauritius or Mali, density and localisation of SNPs in the *S. haematobium* reference genome are shown in 2 kb non-overlapping regions, with each point coloured by the species with the closest nucleotide sequence homology. For each sample, SNP-rich regions (light green blocks) of which $> 20\%$ resembled a genomic reference other than *S. haematobium* are labelled (i-xi). (TIFF)

**S1 Dataset. Variant call format (VCF) file, including single nucleotide polymorphisms (SNPs) reported in an individual male *Schistosoma haematobium* worm from Mali (Mi).** (ZIP)

**S2 Dataset. Variant call format (VCF) file, including single nucleotide polymorphisms (SNPs) reported in an individual male *Schistosoma haematobium* worm from Mauritius (Ms).** (ZIP)

**S3 Dataset. Variant call format (VCF) file, including single nucleotide polymorphisms (SNPs) reported in an individual male *Schistosoma haematobium* worm from Senegal (S1).** (ZIP)

**S4 Dataset. Variant call format (VCF) file, including single nucleotide polymorphisms (SNPs) reported in an individual male *Schistosoma haematobium* worm from Zambia**

**(Z1).**
(ZIP)

**S1 Table. New *Schistosoma haematobium* sequence data produced in this study and linked to NCBI sequence read archive submission details.**
(XLSX)

**S2 Table. Synteny and contiguity of the *Schistosoma haematobium* reference genome (Shae.V3), compared with that of other schistosomes.**
(XLSX)

**S3 Table. Orthologs of *Schistosoma haematobium* in *S. mansoni*, *S. japonicum* and *S. bovis*** inferred using OrthoFinder.
(XLSX)

**S4 Table. Transcription levels for *Schistosoma haematobium* genes, determined using StringTie2.**
(XLSX)

**S5 Table. Annotation of inferred *Schistosoma haematobium* proteins using InterProScan.**
(XLSX)

**S6 Table. Annotation of inferred *Schistosoma haematobium* proteins based on matches to the EggNOG database.**
(XLSX)

**S7 Table. Kyoto Encyclopedia of Genes and Genomes (KEGG) orthology, pathway annotation, BRITE and enzyme classification for *Schistosoma haematobium* proteins.**
(XLSX)

**S8 Table. Functional annotation of the five most highly transcribed sequences in seven key developmental stages of *Schistosoma haematobium*.**
(XLSX)

**S9 Table. KEGG terms significantly enriched (q-value < 0.05) in clusters of distinct transcription profiles for *Schistosoma haematobium*.**
(XLSX)

**S10 Table. Differentially transcribed isoforms in adult female *Schistosoma haematobium*, compared with adult males.** Fold change, q-value and TPM (transcript per million) for each library are shown.
(XLSX)

**S11 Table. Differentially transcribed isoforms in adult male *Schistosoma haematobium*, compared with adult females.** Fold change, q-value and TPM (transcript per million) for each library are shown.
(XLSX)

**S12 Table. KEGG terms significantly enriched (q-value < 0.05) among transcripts differentially expressed in adult male or female *Schistosoma haematobium*.**
(XLSX)

**S13 Table. Modelling of the predicted structures of seven *Schistosoma haematobium* omega-1 homologs using AlphaFold and alignment employing TM-align.**
(XLSX)

**S14 Table. Information on collection site, host, year and Natural History Museum reference code for four isolates of adult male *Schistosoma haematobium* from Africa.**
(XLSX)

## Acknowledgments

Parasite materials (cercariae, adult male and female worms of *Schistosoma haematobium*, Egyptian strain) were provided by the NIAID Schistosomiasis Resource Center for distribution through BEI Resources, NIAID, NIH. We are grateful to Ashling Charles from the DNA Zoo Australia team for support in the routine processing of data.

## Author Contributions

**Conceptualization:** Andreas J. Stroehlein, Robin B. Gasser, Neil D. Young.

**Data curation:** Andreas J. Stroehlein, Pasi K. Korhonen, Aidan M. Emery, Neil D. Young.

**Formal analysis:** Andreas J. Stroehlein, Parwinder Kaur, Olga Dudchenko, Neil D. Young.

**Funding acquisition:** Pasi K. Korhonen, Bicheng Yang, Bill C. H. Chang, Robin B. Gasser, Neil D. Young.

**Investigation:** Andreas J. Stroehlein, Louis-Albert Tchuem Tchuenté, J. Russell Stothard, Parwinder Kaur, David Rollinson, Robin B. Gasser, Neil D. Young.

**Methodology:** Andreas J. Stroehlein, Stuart A. Ralph, Parwinder Kaur, Olga Dudchenko, Bonnie L. Webster, Robin B. Gasser, Neil D. Young.

**Project administration:** Neil D. Young.

**Resources:** Andreas J. Stroehlein, Pasi K. Korhonen, Margaret Mentink-Kane, Hong You, Donald P. McManus, Louis-Albert Tchuem Tchuenté, J. Russell Stothard, Olga Dudchenko, Bicheng Yang, Aidan M. Emery, Bonnie L. Webster, Paul J. Brindley, David Rollinson, Robin B. Gasser.

**Software:** Andreas J. Stroehlein.

**Supervision:** Robin B. Gasser, Neil D. Young.

**Validation:** Andreas J. Stroehlein.

**Visualization:** Andreas J. Stroehlein, Neil D. Young.

**Writing – original draft:** Andreas J. Stroehlein, Neil D. Young.

**Writing – review & editing:** Andreas J. Stroehlein, V. Vern Lee, Stuart A. Ralph, Margaret Mentink-Kane, Hong You, Donald P. McManus, Louis-Albert Tchuem Tchuenté, J. Russell Stothard, Parwinder Kaur, Erez Lieberman Aiden, Huanming Yang, Aidan M. Emery, Bonnie L. Webster, Paul J. Brindley, David Rollinson, Bill C. H. Chang, Robin B. Gasser, Neil D. Young.

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
