## [Decision Letter · Decision Letter 0]

6 Dec 2021

Dear Dr. Young,

Thank you very much for submitting your manuscript "Chromosome-level genome of Schistosoma haematobium underpins genome-wide explorations of molecular variation" for consideration at PLOS Pathogens. As with all papers reviewed by the journal, your manuscript was reviewed by members of the editorial board and by several independent reviewers. In light of the reviews (below this email), we would like to invite the resubmission of a significantly-revised version that takes into account the reviewers' comments.

I am returning your manuscript with three reviews. The reviewers came to different conclusions about the scope of potential revisions, as you will see. After reading the reviews and looking at the manuscript, I recommend Major Revision. While there is no single shared major contention, the reviewers each point to a number of both minor and significant issues that require greater clarification and additional analyses. I am sorry I cannot be more positive at the moment, however we are looking forward to receiving your revision in response to these reviews. I expect the manuscript will be suitable for resubmission with no additional wet lab experiments.

We cannot make any decision about publication until we have seen the revised manuscript and your response to the reviewers' comments. Your revised manuscript is also likely to be sent to reviewers for further evaluation.

Sincerely,

Mostafa Zamanian, Ph.D.

Guest Editor

PLOS Pathogens

James Collins III

Section Editor

PLOS Pathogens

Kasturi Haldar

Editor-in-Chief

PLOS Pathogens

orcid.org/0000-0001-5065-158X

Michael Malim

Editor-in-Chief

PLOS Pathogens

orcid.org/0000-0002-7699-2064

I am returning your manuscript with three reviews. The reviewers came to different conclusions about the scope of potential revisions, as you will see. After reading the reviews and looking at the manuscript, I recommend Major Revision. While there is no single great major point of contention, the reviewers each point to a number of both minor and significant issues that require greater clarification and additional analyses. I am sorry I cannot be more positive at the moment, however we are looking forward to receiving your revision in response to these reviews. I expect the manuscript will be suitable for resubmission with no additional wet lab experiments.

Reviewer's Responses to Questions

**Part I - Summary**

Reviewer #1: The manuscript by Stroehlein et al. provides the third version of genome of Schistosoma haematobium (Shae. V3, from the Egyptian strain), which is also the first chromosome-level assembled genome. Using the Shae.V3, the investigators identified the SNPs of four males collected from four geographic locations in Africa. Furthermore, they compared gene expression of the parasites at different developmental stages, including eggs from human urines and hamster livers as well as male and females, and revealed key genes that are predominately expressed in the different developmental stages or sexes. The study provides basic genomic information, improves our understanding of S. haematobium biology, and will have an important impact to research of schistosomiasis, particularly urogenital schistosomiasis, the most prevalent form of schistosomiasis in the world. As S. haematobium is the causative pathogen of urogenital schistosomiasis, the study is well suitable for PLoS Pathogens.

Reviewer #2: Stroehlein and colleagues describe the generation and analysis of a chromosome-scale assembly of Schistosoma haematobium, an important pathogen of humans. This genome assembly represents a significant improvement on existing resources available for S. haematobium, and now allows novel insight into genome evolution via introgression and sex-specific transcription as the manuscript explores.

Overall, it is a well written manuscript that describes a useful genomic resource that is needed for schistosomiasis communities focused on the genetic epidemiology of the parasite and development of novel control interventions. I did find some areas lacking in a rationale, and/or specific details, that if improved, would help both a broad-interest reader of the journal and also someone interested in the intricate parts of the analysis. I have made some suggestions to this effect below.

For full disclosure, I discussed this manuscript with a PhD student in our group, Duncan Berger.

In my opinion, this manuscript is a good fit for PLoS Pathogens, and I would be very happy to see a revised version.

Kind regards,

Stephen Doyle

Wellcome Sanger Institute

Abstract

- It is quite background heavy, and doesn’t focus much at all on what was done and what was found. These two aspects could be expanded on.

Introduction

- Check the reference formatting. Ie. {Knopp, 2019 #6919;Aula, 2021 #6575}.

- “these studies usually…limited number of nuclear genetic markers” – perhaps this is a little unfair on the recent use of genomics involving S. haematobium, for example:

o Platt et al 2019 MBE

o Oey et al 2019 PLoS Pathogens

o Rey et al 2021 PLoS Pathogens – you do cite this, but before you state the limitation of no genomics

Results

1. Reference genome

- I like figure 1. It could be helpful to mention the syntenic blocks are five or more single copy orthologs.

- Following on, differences in block length are not obvious at the scale presented on the CIRCOS plots. Are you able to comment on the distribution of block lengths? Are there parts of genomes that are more conserved, relative to other regions, based on conservation of block length?

- Figure shows “ZW chromosome” – these are obviously two chromosomes – Z and W - not one as indicated. Were you able to separate & define them in the assembly? This deserves some mention, especially given the efforts to do this by Buddenborg et al 2021 bioRxiv for S. mansoni.

2. Gene models and annotations

- What proportion of gene models were transferred by LiftOff?

- Table 2 – can you add the accession numbers for the assemblies? The assembly and annotation of S. mansoni at least has changed over time, so would be good to be clear which is being analysed.

3. Molecular variation

- “distinct geographical locations” – perhaps best describe where they are and how many worms.

- Following on from this, it seems like only 4 samples were compared. There are other WGS datasets available for S. haematobium from other populations (eg. PRJNA561522) . Did you consider using some of these datasets?

- “laboratory reference strain,” – where is the reference strain from?

- “moderate or high impact” – there is huge difference in the potential effect of these two – perhaps best to state these numbers explicitly.

- “SNP-dense regions” – this description of the results is a little shallow. How is these regions defined, specifically? There is some implication that these represent introgression regions (at least in the discussion: “unique genomic signatures matching species other than S.haem…”) – if so, these should be more formally described.

- Supplementary Fig 1 is almost impossible to interpret. Could be better in landscape format perhaps?

4. Variation in the transcriptome…

- There is a focus on the functional annotation/gene descript of the top 1% of genes, but is there an actual enrichment of these gene sets in the top 1%, or are they just present?

5. Differential isoform usage

- I don’t think Fig 3a communicates that there are distinct isoforms in two or more stages. Just simply looks like stage-specific expression.

- The methods are very light on this, and in fact, don’t mention any specific analysis for quantifying differential isoform usage. Did you consider formally describing this – there are a number of tools available, and some precedent, for example, for H. contortus in Doyle et al 2020 CommBiol.

6. Genes with similar transcription profiles

- “between 817 and 4301 transcripts” - which is it? How many in transcripts in total could be assigned to at least one of the clusters?

7. Sex-linked transcription

- “genes located on the ZW chromosome” – similar to the question above, did you defined Z-specific and W-specific genes? Given you are talking about male and female specific transcription, this is pretty important.

- Similarly, genes present in the pseudo-autosomal regions of Z and W may also be interesting, particularly if you saw sex-specific expression. Did you see this at all?

8. Differential transcription in eggs

- The results could use a single introductory sentence describing the rationale for doing this. You have it in the discussion, however, I think it is needed here too.

- Similarly, there are clearly results presented in the discussion that should be moved back into results, that might help round this section out a bit more.

Discussion

- “well-defined laboratory (BRI) line” – perhaps remind the reader that it is Egyptian, as it becomes more relevant to describing African diversity.

- As a follow up, Biomedical Research Institute is never actually abbreviated as BRI, so that could perhaps be clarified.

- Introgression / “unique genomic signatures” – this is potentially really interesting, esp given the background, however, I found the analysis quite limited. There are a number of tools specifically focused on identifying introgression events based on phylogenetic incongruence, dating them etc. A lot in the Helionchus literature for example. The methods described here represent an interesting, perhaps novel approach to this question, but it is not clear how this approach compares to more established methods.

- “this scaffold represents a female-specific portion of the W chromosome” – can you check this and others with female WGS libraries? To what degree did you try and define W-specific sequences – this is not clear, but is relevant to some analyses and should be described.

- “The proposal that variation in transcription levels in eggs…” – this is not clear to me. Who is making this proposal?

- As a follow up, this sentence doesn’t make sense to me. States the differences in transcription between hamster and human relate to interaction between parasite and host and/or pathology – do you mean differences in transcription relate to differences in host-interetions and/or differences in pathology of infection? If so, this could be made clearer for a naïve reader.

- AlphaFold results should be in the results, and results shown somewhere.

- “From a technical perspective, the short-read sequencing…” – it reads like you have done this for the first time, which is not technically correct. I suggest tone this down.

Methods

- For all sequencing, including the use of older, publicly available data – was BRI used? Not really clear.

- HiC – what method was used? Was there any manual curation of the assembly based on the contact map information?

- Synteny analysis – “linking single-copy orthologs in a pairwise manner” – what does this mean exactly?

Code

- Great that the code for the analysis and recreating of the figures is made available on gitlab, however, I can’t seem to access it. Could the authors please check it is visible.

Data availability

- I can’t see the V3 genome in ENA yet under the project ID provided. Can you confirm the genome and annotation have been submitted somewhere.

Reviewer #3: In the submitted manuscript, the authors present an updated version of the genome for Schistosoma haematobium. The associated genome annotation has also been updated. The authors describe their population genomic analysis of S. haematobium from four additional geographical locations. The authors also describe an in depth transcriptional analysis, where they look at gene expression across life-cycle stages, between sexes, and how isoforms of individual genes change.

Strengths & Novelty

This is the third version of the S. haematobium genome. The improvement to the genome continuity is impressive: a 10-fold increase in N50 and 98% of the assembly present on just 8 scaffolds. The increase is BUSCO single-copy genes is small but demonstrates a notable jump in accuracy. Overall, these metrics demonstrate the value of undertaking Hi-C sequencing.

To the best of my knowledge, two aspects of the gene expression analyses were novel: differential isoform usage and the sex-linked expression.

There are several new sequencing datasets: Hi-C (gDNA); Nanopore long-reads (gDNA); four distinct populations (gDNA); adult worms, including sexed worms (RNA); cercariae (RNA); and schistosomules (RNA). These data will be partially responsible for the important improvement in the accuracy of the annotation of the genome (gene models). From my reading of the Methods section, the adult worms were collected in three or six replicates. It appears that the other RNA datasets were without replication (see later in this review)

In the main, the level of detail in the Methods would allow others to replicate the study.

Weakness

While the assembly and annotation are notable improvements over previous versions, the manuscript felt like an announcement of an incremental update and presentation of a resource. I finished reading the manuscript and wondered what major new understanding of S. haematobium biology was discovered. Analysis of the RNA-Seq data is a major feature of the results, and it is difficult to tease out what are the novel discoveries. For example, “[i]n the sporocyst stage, unique transcripts encoded… venom allergen-like” [proteins].” However, in Young et al. 2012, venom allergen-like proteins “ES proteins in the egg.” And, in figure 3 of the earlier paper, these proteins are highly expressed in the egg compared to adults. The findings of the submitted manuscript are not necessarily incongruent with the earlier paper, but proper context is required.

The section of molecular variation between four locations was purely descriptive. In the Discussion, the authors call the observed variation ‘substantial’, however, they give no context with respect to other Schistosoma species. Further, I would like them to consider the work of Gower et al 2013 (http://dx.doi.org/10.1016/j.actatropica.2012.09.014) would investigated S. haematobium population structure.

Due to the journal’s requirement for a quick turn around time for the review, I did not have time to cross reference every gene mentioned with previous papers presenting -omic datasets for S. haematobium. However, I consider it important that the authors present what is novel about the current study, especially around the transcriptomics.

**Part II – Major Issues: Key Experiments Required for Acceptance**

Reviewer #1: (No Response)

Reviewer #2: (No Response)

Reviewer #3: The population genomic analysis is presented as an UpSet plot and piecharts. This lacks the sophistication that has become standard in population geniomics. I strongly recommend that the authors use the program STRUCTURE (https://web.stanford.edu/group/pritchardlab/structure.html) or another robust approach.

Differential gene expression features prominently in the submitted manuscript. Not all the transcriptomics data are available in triplicate. This means it is not possible to conduct statistically robust comparisons (doi.org/10.1261/rna.053959.115). Either the authors need to generate appropriate replication datasets for all stages under consideration or restrict their analysis to only those datasets done in triplicate.

**Part III – Minor Issues: Editorial and Data Presentation Modifications**

Reviewer #1: 1) In the section of Introduction. drug can sometimes be ineffective 9-11, and treatment alone does not achieve sustainable control of this disease in endemic countries {Knopp, 2019; #6919;Aula, 2021 #6575}. The drug may be ineffective for juvenile schistosomes, but I do not understand what “Sometimes” means in the sentence and also have no idea about the information in parentheses { }.

2) Three versions of S. haematobium were generated from the same group in collaboration with different investigators. I understand that the third version (chromosome level assembly) was produced using a combination of Hi-C sequencing and other data. It is unclear which data are unpublished or new and which were published or used in the previous publications. Also in the section of Result, it stated that Hi-C and long-read data were used. In the section of Materials and Methods, it said short Illumina reads were also used.

3) In the section of results: A comparison of genome Shae.V3 with a previous assembly for S. haematobium (i.e. Shae.V2 with 130 scaffolds)23 (Table 1; Fig. 1d) showed that Shae.V3 is substantially more contiguous. I expected Shae V3 should have a smaller number of scaffolds as compared to Shae V2. Here it said the number of scaffolds of Shae V2 is 130 whereas in Shae. V3, the number is 163. Table 1 however indicates Shav V2 has 666 scaffolds. Please clarify it.

4) A number of genomes from different Schistosoma species are compared. It is unclear which version of genome for a given species are used in Table 1 as some species have more than one published version of genome sequences.

5) It is interest to know what re-arrangements are, although supplementary Table 2 provides such information. Using diagrams to illustrate the re-arrangements is recommended.

6) This is a genome-level assembly. In theory, it should be assembled from 8 scaffolds. It is understandable that repeated sequences make it difficult to assemble chromosome scale scaffolds. But I think that the first 8 big scaffolds should represent most majority of the genome (i.e., 8 chromosomes). Such important information is not described in the paper.

7) For the BUSCO analysis, it stated that the number of core genes within the Eukaryota dataset in Table 1 is 255, but the number in Table 2 is 954. I assume 954 is for Metazoa, which however is also different from Eukaryota.

8) How to distinguish fixed SNPs (i.e., homozygous) from non-fixed SNPs?

9) RNAseq: Description of materials and methods is incomplete and unclear throughout the paper. The description was emphasized on how to use different softwares. How to properly handle biological materials is critical for obtaining reliable RNAseq data for subsequent analyses because gene expression is affected by many environmental factors, such as temperature of preserving or transporting samples, degradation of tissues or RNA samples, etc. Also, the data showed expression of the two sexes, but the materials said 50 worm pairs, in addition single sex samples, were used. Following questions should be clearly addressed in the paper. What biological replicates and/or technical replicates are for each sample? Where and how RNA extraction (for example, eggs from human urines) were done? How to transfer live tissue samples or RNAs to lab, etc.

Reviewer #2: (No Response)

Reviewer #3: The SRA data for this paper remain ‘not public’. When released will the data include the sequencing runs for the four populations from geographical locations? I ask because, in the methods, I could not find specific run accessions for these in the paper.

PLOS authors have the option to publish the peer review history of their article (what does this mean?). If published, this will include your full peer review and any attached files.

Reviewer #1: No

Reviewer #2: **Yes: **Stephen R Doyle

Reviewer #3: No
---

## [Editor Report · Decision Letter 1]

19 Jan 2022

Dear Dr. Young,

We are pleased to inform you that your manuscript 'Chromosome-level genome of Schistosoma haematobium underpins genome-wide explorations of molecular variation' has been provisionally accepted for publication in PLOS Pathogens.

Best regards,

Mostafa Zamanian, Ph.D.

Guest Editor

PLOS Pathogens

James Collins III

Section Editor

PLOS Pathogens

Kasturi Haldar

Editor-in-Chief

PLOS Pathogens

orcid.org/0000-0001-5065-158X

Michael Malim

Editor-in-Chief

PLOS Pathogens

orcid.org/0000-0002-7699-2064
---

## [Editor Report · Acceptance letter]

31 Jan 2022

Dear Dr. Young,

We are delighted to inform you that your manuscript, "Chromosome-level genome of *Schistosoma haematobium* underpins genome-wide explorations of molecular variation," has been formally accepted for publication in PLOS Pathogens.

Best regards,

Kasturi Haldar

Editor-in-Chief

PLOS Pathogens

orcid.org/0000-0001-5065-158X

Michael Malim

Editor-in-Chief

PLOS Pathogens

orcid.org/0000-0002-7699-2064